# HCFPN: Hierarchical Contextual Feature-Preserved Network for Remote Sensing Scene Classification

Jingwen Yuan  and Shugen Wang *

School of Remote Sensing and Information Engineering, Wuhan University, Wuhan 430072, China
* Correspondence: wangsg@whu.edu.cn

**Abstract:** Convolutional neural networks (CNNs) have made significant advances in remote sensing scene classification (RSSC) in recent years. Nevertheless, the limitations of the receptive field cause CNNs to suffer from a disadvantage in capturing contextual information. To address this issue, vision transformer (ViT), a novel model that has piqued the interest of academics, is used to extract latent contextual information in remote sensing scene classification. However, when confronted with the challenges of large-scale variations and high interclass similarity in scene classification images, the original ViT has the drawback of ignoring important local features, thereby causing the model's performance to degrade. Consequently, we propose the hierarchical contextual feature-preserved network (HCFPN) by combining the advantages of CNNs and ViT. First, a hierarchical feature extraction module based on ResNet-34 is utilized to acquire the multilevel convolutional features and high-level semantic features. Second, a contextual feature-preserved module takes advantage of the first two multilevel features to capture abundant long-term contextual features. Then, the captured long-term contextual features are utilized for multiheaded cross-level attention computing to aggregate and explore the correlation of multilevel features. Finally, the multiheaded cross-level attention score and high-level semantic features are classified. Then, a category score average module is proposed to fuse the classification results, whereas a label smoothing approach is utilized prior to calculating the loss to produce discriminative scene representation. In addition, we conduct extensive experiments on two publicly available RSSC datasets. Our proposed HCPFN outperforms most state-of-the-art approaches.

**Keywords:** RSSC; hierarchical multiheaded attention; label smoothing



## 1. Introduction

Remote sensing scene classification (RSSC), which is highly significant in Earth observation applications, such as urban planning, land cover classification, and geographic object detection [1–5], is generally based on high-resolution remote sensing (HRRS) images. As Earth observation technology advances, the amount of HRRS images is continuously rising. Thus, taking full advantage of the growing number of HRRS images for intelligent Earth exploration is important [6,7]. As a result, comprehending massive and complicated HRRS images has become a critical and challenging task. Based on its content, the RSSC seeks to categorize the given remote sensing imagery into predefined semantic categories. Many comprehensive academic studies [8–13] on RSSC have been conducted in the last few decades. Initially, most RSSC techniques focused primarily on human-engineering features, including scale-invariant feature transformation (SIFT) [14], histogram of oriented gradients (HOG) [15], and color histogram (CH) [16]. Nevertheless, the handcrafted features have limited representational capacities because they are unable to fully capture the semantic content of HRRS scenes.

As a result of the development of convolutional neural networks (CNNs), Numerous CNN-based approaches for RSSC have been developed [17–22]. CNNs have a strong capacity to extract high-level abstract characteristics given that they are designed to resemble the

visual system of the human brain. As a consequence, the CNN-based algorithms achieve excellent results in RSSC. There have been several general-purpose CNNs proposed, including VGG-Net [23] and residual network (ResNet) [24]. Furthermore, numerous graph neural networks or CNNs oriented to remote sensing have been proposed [25–29], which perform excellently in RSSC. Despite the fact that the performance of CNN-based methods in scene classification has significantly improved, long-term contextual information concealed in remote sensing scenes cannot be effectively explored. CNN features mainly reflect information from local regions while ignoring long-term contextual dependencies between local regions. As demonstrated in Figure 1, this condition is vital for defining semantic categories of different remote sensing scenes. The nonlocal method [30] and the conformer strategy [30] are two common approaches by scholars to solve these issues. Transformer [31] stands out among the alternatives because of its capacity to learn long-term contextual information. Thus, various RSSC models based on vision transformers have been presented [32–35].

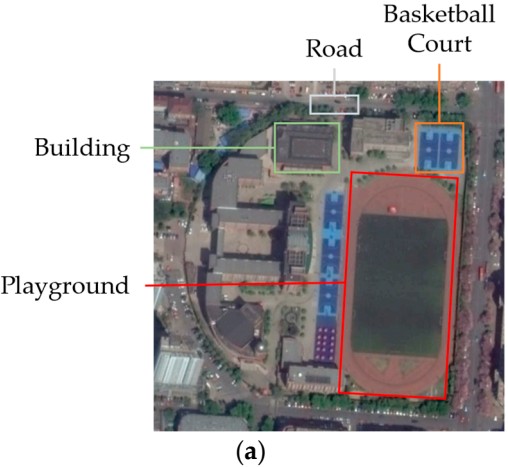
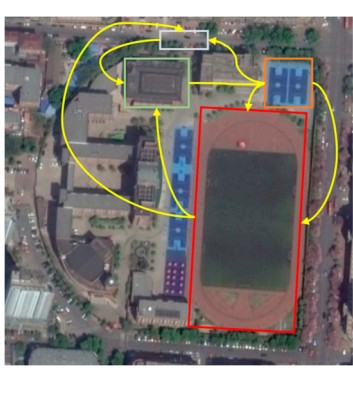

(**a**)    (**b**)

**Figure 1.** (**a**) Local visual feature and (**b**) long-term contextual feature are selected from the "School" category. The long-range contextual information is not considered by CNNs, which specialize at extracting local characteristics. (**a**,**b**) show many distinct land covers, such as "Playground," "Building," "Road," and "Basketball Court." If the model only concentrates on those local regions, the "School" scene could be mistaken for other scenes with comparable ground objects. Thus, to accurately classify this scene as "School," local regions and their long-term contextual dependencies, as indicated by the yellow arrows, should be taken into account by the expected model.

Nevertheless, the transformer model can still be improved in terms of interpreting HRRS scenes. First, although the original ViT estimates the contextual dependencies between patches created by image cropping, it has limitations when it comes to investigating the local structural information of HRRS scenes with intricate geometrical structures and spatial patterns. Second, most transformer-based approaches ignore multilevel local features and are limited to extracting contextual information at a single scale. Following the discussion above, two questions logically arise: (1) whether it is feasible to explore local information and long-term contextual information while combining the advantages of CNNs and ViT; (2) efficiently fusing high-level semantic features of CNN and multilevel contextual features to generate discriminative feature representations.

Following the preceding analysis, we propose a new method, namely, hierarchical contextual feature-preserved network (HCFPN) for remote sensing scene classification. First, we utilize a hierarchical feature extraction module (HFEM) to extract multilevel convolutional features and high-level semantic features from HRRS scenes. Second, a contextual feature preserved module (CFPM) with a multiheaded cross-level attention is proposed to capture multilevel long-term contextual features hidden in HRRS scenes. The scattered multilevel long-term contextual features can be aggregated to form a more

discriminative representation and facilitate subsequent classification by calculating the multiheaded cross-level attention. Third, the classification outcomes of the high-level semantic features and the multilevel long-term contextual features are fused using a category score average module (CSAM). In addition, we present the label smoothed cross-entropy loss to strengthen the model's classification performance while avoiding the overfitting issue in deep learning.

The following is a summary of this manuscript's significant contributions:

1. To efficiently integrate the benefits of CNN and ViT to extract both local high-level semantic features and global contextual features, an RSSC network called HCFPN is proposed. In addition, HRRS scenes can be comprehensively exploited and aggregated for the local high-level semantic features and global contextual features.
2. To describe the correlations of multilevel convolutional features and merge them to generate a more discriminative representation, the global long-term contextual features of the multilevel convolutional features are captured through a multiheaded self-attention, and then the correlations between them are further explored through multiheaded cross-level attention.
3. Extensive experiments are carried out on two public benchmark datasets, and the superior outcomes illustrate that our proposed HCFPN works effectively in RSSC.

The remaining manuscript is structured as follows. Section 2 provides an overview of the relevant academic research. Section 3 provides detailed information about the proposed HCFPN. Section 4 presents the experimental results of the HCFPN, utilizing two well-known HRRS scene classification datasets. Section 5 analyzes the rationale for the viability of our proposed method. Finally, Section 6 describes the conclusions.

## 2. Related Works

### 2.1. CNN-Based Methods for RSSC

A variety of scene classification approaches based on CNNs have evolved to learn good feature representations by applying various strategies of exploiting CNNs. Existing CNN-based approaches may be categorized into three groups depending on the architecture of the models: single-branch, multi-branch, and generative adversarial framework.

The scene classification networks in the first group are single-branch structures, indicating that they have only one sample input entrance. Li et al. [36] applied the transfer learning paradigm to the training process of CNNs, learning robust visual feature representations and enhancing the scene classification performance in the case of limited labeled samples. Meanwhile, an attribute-cooperated CNN [37], which attributes learning to distinguish visually fine-grained categories, was proposed. In addition, aggregating multilayer features is a commonly used strategy for RSSC. A convolutional feature encoding module as well as a progressive aggregation strategy were included in an end-to-end feature aggregation CNN (FACNN) [38] that was developed to learn discriminative scene representation. This allows the intermediate features to be aggregated while fully leveraging the semantic label information. To obtain a global discriminative feature representation for HRRS scenes, the low-level and middle-level convolutional features were encoded by a vector of a locally aggregated descriptor (VLAD) in [39]. Then, an encoded mixed-resolution representation strategy was proposed to concatenate all the global features.

In the second group, the scene classification networks contain a multibranch structure that allows them to deeply explore the inter-class and intra-class interactions between HRRS scenes. The Siamese network [40], which combines two weight-sharing CNNs, is a common example of a multibranch CNN. In addition, some particular objective functions were created in the Siamese network to achieve content interpretation of HRRS scenes. In [41], deep structural metric learning and the Siamese network were integrated to extract features and construct a diversity-promoting prior, which improved the classification performance of the model. Liu et al. [42] proposed a Siamese network for RSSC that integrates identification and verification models to learn discriminative feature representations. Compensating for the shortcomings of the verification model, the identification model evaluated

the correlation of all the images in a dataset. The intra-class distance was decreased by the verification model, while the inter-class distance was increased.

In the third group, the scene classification networks are based on a generative adversarial framework that generates real-like data and is based on a minmax two-player game. To produce high-resolution annotated samples, [43] developed a generative adversarial network (GAN)-based remote sensing image generation (GAN-RSIGM) approach. The Wasserstein distance was used by GAN-RSIGM to promote the distribution of the generator near the distribution of real data. A supervised progressive growing GAN (SPG-GAN) [44] was proposed to generate HRRS scenes with labeled categories. In addition, SPG-GAN adopted a progressive growing sample generation strategy to enhance the spatial details of the generated samples. A perturbation-seeking GAN (PSGAN) [20] was presented for RSSC to boost the defense against unknown threats. PSGAN initially trained the model, utilizing the adversarial samples with perturbations rather than clean samples. Moreover, a scale factor was proposed to make a trade-off between PSGAN optimization and the diversity of the rebuilt samples.

### 2.2. Attention-Based Methods for RSSC

In complicated HRRS scenes, highlighting critical information and capturing contextual dependence information remain two pressing difficulties. To alleviate them, visual attention mechanisms have become interesting to scholars. As a result, several attention-based approaches for RSSC have been proposed; they can be classified as either conventional-attention-based or transformer-based methods.

The models based on conventional attention are intended to extract saliency areas from HRRS scenes for classification tasks. In [45], an end-to-end attention recurrent convolutional network (ARCNet) was proposed to focus selectively on particular crucial regions or locations, consequently eliminating the redundant information and improving the model's classification performance. Zhao et al. [46] proposed an enhanced attention module (EAM) for RSSC to extract more discriminative representations. The EAM was constructed with simple and effective enhanced spatial attention and enhanced spectral attention modules. In the attention consistent network (ACNet) [47], a parallel attention mechanism was designed to focus on the local features from spatial and spectral dimensions. In addition, an attention consistent module was developed to unify various types of attention maps to increase intra-class distance while decreasing inter-class distance.

The second group of models is mainly constructed by the transformer. The transformer initially reigned supreme in the natural image processing field owing to its ability to capture contextual information concealed in images [48–50]. The first attempt of vision transformer [48] was in the field of computer vision. Prior to performing the classification task, it transforms natural pictures into sequences of image patches and inserts their absolute location information into these image patches. In the field of RSSC, several transformer-based methods have been proposed. A spatial-channel feature preserving ViT (SCViT) was proposed in [51], which takes into account the abundance of geometric information in high-resolution images as well as the contribution of the various channels included in the classification token. A transformer-driven semantic relation inference network with semantic sensitive and semantic relation-building module was proposed in [52]. The semantic sensitive module detected the pivotal semantic attentional regions in the feature map, whereas the semantic relation-building module predicted final outcomes using label relation inference from the semantic sensitive module outputs.

## 3. Proposed Method

The general structure of the proposed HCFPN is illustrated in Figure 2, which comprises a hierarchical feature extraction module, a contextual feature preserved module, and a category score average module. HFEM is designed to extract hierarchical convolutional features as well as local advanced semantic features from HRRS scenes. The purpose of CFPM is to acquire long-term contextual information from HRRS scenes at a different scale

by capturing the long-term contextual relationships between local convolutional areas. In addition, CSAM is intended to combine advanced semantic features and long-term contextual features to provide discriminative feature representation for RSSC. Then, we provide a thorough description of each module.

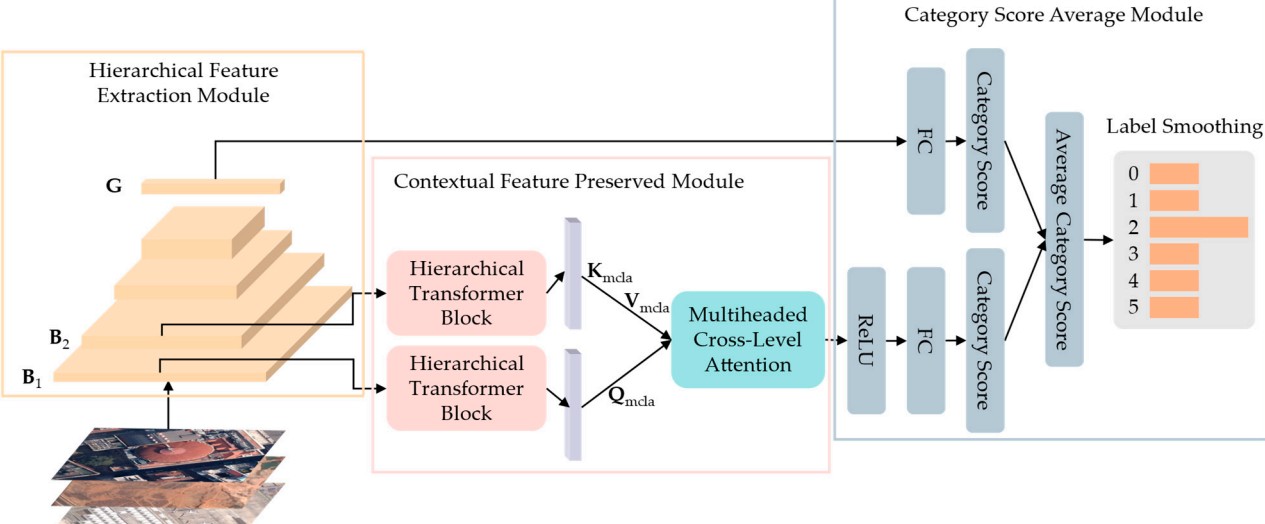

**Figure 2.** Overall framework of the proposed HCFPN, which comprises a hierarchical feature extraction module, a contextual feature preserved module, and a category score average module. $\mathbf{B}_1$, $\mathbf{B}_2$, and $\mathbf{G}$ represents low-level convolutional features, mid-level convolutional features, and high-level semantic representations. $\mathbf{Q}_{mcla}$, $\mathbf{K}_{mcla}$, and $\mathbf{V}_{mcla}$ are the queries, keys, and values involved in the multihead cross-level attention calculation, respectively.

### 3.1. Hierarchical Feature Extraction Module

The hierarchical feature extractor is based on ResNet34, a widely used CNN consisting of four residual blocks, a global average pooling layer, and a fully connected (FC) layer. The residual blocks focus on extracting local spatial information, and the significance of global average pooling is that it helps to regularize the entire network structure to avoid overfitting. Furthermore, the classification score is generated using a global feature vector from the final FC layer [53]. The residual blocks of different depths focus on extracting different hierarchical information. Shallow residual blocks usually extract the low-level features, such as color, texture, and shape. Deeper residual blocks focus on extracting high-level semantic representations with abundant combinatorial information, which are important in solving classification problems. Considering the complex contextual information of HRRS scenes, the features at various levels contribute to the RSSC task. As shown in Figure 2, to prevent the information loss of shallow features, we select the output of the first two residual blocks to construct low-level and mid-level convolutional features that will help in the generation of more discriminative features. For clarity, they are designated as $\mathbf{B}_1 \in \mathbb{R}^{C_1 \times H_1 \times W_1}$ and $\mathbf{B}_2 \in \mathbb{R}^{C_2 \times H_2 \times W_2}$, where $C_1$ and $C_2$ are the number of channels and $(H_1 \times W_1)$ and $(H_2 \times W_2)$ are the resolutions of the low-level and mid-level features, respectively. Moreover, $\mathbf{G} \in \mathbb{R}^{C_3}$ represents the aggregated high-level semantic representations generated by the global average pooling layer.

### 3.2. Contextual Feature Preserved Module

Some detailed information may be lost when shallow features (e.g., $\mathbf{B}_1$ and $\mathbf{B}_2$) are passed forward to deep residual blocks. At the same time, the receptive field of the convolution process is limited by the size of the convolution kernel, which might cause CNNs to focus primarily on local information and ignore the global dependence of image blocks. To completely comprehend the contents of HRRS scenes, we propose CFPM, which

is based on the HTB, to extract contextual information. To describe the HTB in detail, we consider $\mathbf{B}_1$.

A 1D sequence of token embeddings is sent into the HTB as input. To deal with 2D shallow feature $\mathbf{B}_1$, we reshape it into a sequence of flattened 2D patches, $\mathbf{B}_1^p \in \mathbb{R}^{N \times (P^2 \cdot C_1)}$, where $N = (H_1 \times W_1)/P^2$ represents the total number of patches and $(P \times P)$ represents the size of each feature patch. Given that the latent features utilized in all HTB's layers are constant to $D$ dimensions, a linear projection, $\mathbf{E} \in \mathbb{R}^{(P^2 \cdot C_1) \times D}$ (Equation (1)), which can be trained, is used to flatten and map the patches to $D$ dimensions. The output of this linear projection is referred as the patch embeddings. A learnable embedding, $\mathbf{B}_{class\_token}$, is prepended for the patch embeddings, similar to BERT's class token, and the image representation, $\mathbf{B}_1' \in \mathbb{R}^{(N+1) \times D}$, is made from its state at the output of the HTB. Furthermore, learnable 1D positional embeddings, $\mathbf{E}_{pos} \in \mathbb{R}^{(N+1) \times D}$, are introduced to the embedded patches to retrain global positional information, which are vital for seeking long-term contextual dependence between image patches.

As shown in Figure 3, an HTB is made up of layers of MSA and MLP blocks [31] that alternate. Layer norm (LN) [54] is used in front of every block, and in the rear, residual connection is utilized. Two FC layers that are activated by a Gaussian error linear unit (GELU) form the MLP blocks. As a result, the HTB output can be calculated as follows:

$$\mathbf{z} = \text{Concat}\left( \mathbf{B}_{class\_token}; \left(\mathbf{B}_1^p\right)^1 \mathbf{E}; \left(\mathbf{B}_1^p\right)^2 \mathbf{E}; \cdots ; \left(\mathbf{B}_1^p\right)^N \mathbf{E} \right) + \mathbf{E}_{pos}, \tag{1}$$

$$\mathbf{z}_{MSA} = \text{MSA}(\text{LN}(\mathbf{z})) + \mathbf{z}, \tag{2}$$

$$\mathbf{B}_1' = \text{LN}(\text{MLP}(\text{LN}(\mathbf{z}_{MSA})) + \mathbf{z}_{MSA}). \tag{3}$$

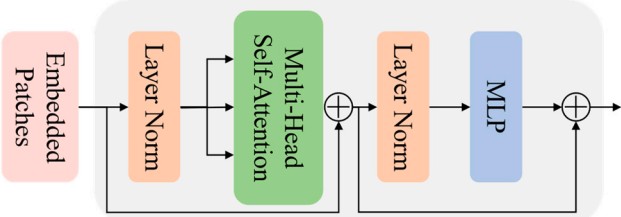

**Figure 3.** Specific details of the hierarchical transformer block (HTB). HTB uses Multiheaded Self-attention (MSA) and Multilayer Perceptron (MLP) Blocks in alternating layers. Each block has a layer norm (LN) in the front, and a residual connection in the back.

The core component of HTB is MSA, which may provide the attention layer output, including information on the encoded representations in multiple subspaces, thereby increasing the model's representational strength. MSA simultaneously uses $h$ self-attention head functions to establish the long-term contextual dependency among token embeddings from different positions while taking the input token embeddings, $\mathbf{z} \in \mathbb{R}^{(N+1) \times D}$, into account. Then, the outputs of the individual heads are concatenated and projected to the ultimate attention values. For input $\mathbf{z}$, the formula for calculating MSA is as follows:

$$\text{MSA}(\mathbf{z}) = \text{Concat}(\text{head}_1, \cdots, \text{head}_h)\mathbf{W}^O, \tag{4}$$

where $\text{head}_i$ represents the $i$th self-attention head function, and $\mathbf{W}^O$ is the parameter matrix of projections. The $\text{head}_i$ is defined as follows:

$$\text{head}_i = \text{Attention}(\mathbf{Q}_i, \mathbf{K}_i, \mathbf{V}_i), \tag{5}$$

$$\mathbf{Q}_i = \mathbf{z}\mathbf{W}_i^Q, \ \mathbf{K}_i = \mathbf{z}\mathbf{W}_i^K, \ \mathbf{V}_i = \mathbf{z}\mathbf{W}_i^V, \tag{6}$$

$$\text{Attention}(\mathbf{Q}_i, \mathbf{K}_i, \mathbf{V}_i) = \text{softamx}\left(\frac{\mathbf{Q}_i \mathbf{K}_i^T}{\sqrt{d_k}}\right)\mathbf{V}_i. \tag{7}$$

where the learned parameter matrices of various linear projections are denoted by $\mathbf{W}_i^Q, \mathbf{W}_i^K$, and $\mathbf{W}_i^V$, and $d_k$ represents the dimension of $\mathbf{K}_i$. The linearly projected queries, keys, and values are denoted as $\mathbf{Q}_i$, $\mathbf{K}_i$, and $\mathbf{V}_i$, respectively; they perform the scaled dot-product attention, Attention$(\cdot, \cdot, \cdot)$, in parallel to yield the $i$th self-attention head's attention score. Finally, the final values are obtained by concatenating these attention scores and linearly projecting them again.

Thus far, the obtained $\mathbf{B}_1' \in \mathbb{R}^{(N+1)\times D}$ and $\mathbf{B}_2' \in \mathbb{R}^{(N+1)\times D}$ contain abundant global contextual knowledge inside HRRS scenes. Although the MSA in HTB may extract global contextual information successfully, it is constrained by single-level convolutional feature. Furthermore, the acquired global contextual knowledge is dispersed and is unfavorable to subsequent classification. To combine the global contextual information from low-level and mid-level convolutional features, multiheaded cross-level attention (MCLA) is proposed to capture the long-term correlation between the low-level and mid-level global contextual features. The long-term correlation between different hierarchical features can be viewed as latent dependence, which can be used to construct a more complete feature representation [19]. Specifically, we construct MCLA directly using the MSA mechanism, which varies from MSA in that MCLA utilizes $\mathbf{B}_1'$ to generate queries and $\mathbf{B}_2'$ to generate keys and values.

$$\mathbf{Q}_{mcla} = \mathbf{B}_1'\mathbf{W}^{Q_{mcla}}, \ \mathbf{K}_{mcla} = \mathbf{B}_2'\mathbf{W}^{K_{mcla}}, \ \mathbf{V}_{mcla} = \mathbf{B}_2'\mathbf{W}^{V_{mcla}}, \tag{8}$$

$$\mathbf{B}_{mcla} = \text{softmax}\left(\frac{\left(\mathbf{B}_1'\mathbf{W}^{Q_{mcla}}\right)\left(\mathbf{B}_2'\mathbf{W}^{K_{mcla}}\right)^T}{\sqrt{d_{k_{mcla}}}}\right)\left(\mathbf{B}_2'\mathbf{W}^{V_{mcla}}\right) \tag{9}$$

where $\mathbf{W}^{Q_{mcla}}$, $\mathbf{W}^{K_{mcla}}$, and $\mathbf{W}^{V_{mcla}}$ are the learned parameter matrices of different linear projections in MCLA, and $d_{k_{mcla}}$ is the dimension of $\left(\mathbf{B}_2'\mathbf{W}^{K_{mcla}}\right)$.

### 3.3. Category Score Average Module

The purpose of CSAM is to average the classification scores computed from high-level semantic features $\mathbf{G}$ and $\mathbf{B}_{mcla}$. Moreover, CSAM utilizes label smoothing to reduce model overfitting and improve the model's generalization capacity. Concretely, we initially take out the class token dimension $\mathbf{B}_{mcla}^{ct} \in \mathbb{R}^D$ in $\mathbf{B}_{mcla}$ and enhance the representational ability of the model with a ReLU activation layer. Then, the $\mathbf{B}_{mcla}^{ct}$ and $\mathbf{G}$ are converted to category scores $\mathbf{S}_{ct}$ and $\mathbf{S}_G$ via an FC layer, respectively. Lastly, $\mathbf{S}_{ct}$ and $\mathbf{S}_G$ are averaged to yield the average contribution with two branches, resulting in a fused category score, $\mathbf{S}$.

$$\mathbf{S} = \frac{\mathbf{S}_{ct} + \mathbf{S}_G}{2}. \tag{10}$$

In this case, the local high-level semantic and global contextual information are merged at the feature level. Thereafter, to classify HRRS scenes, a softmax function is utilized.

Instead of using the standard cross-entropy loss, we utilize the label smoothed cross-entropy loss as the loss function. The label smoothing approach, which creates soft labels by applying a weighted average between uniform distribution and hard label, is a useful regularization technique to lessen the overfitting issue in deep learning. Assuming that the label obtained by softmax function is $\mathbf{y}_S$, the smoothed label, $\mathbf{y}_S^{LS}$, equals the following:

$$\mathbf{y}_S^{LS} = \mathbf{y}_S(1 - \alpha) + \alpha/M, \tag{11}$$

where $\alpha$ is the smoothing parameter and $M$ is the number of categories. At this point, the formula of the label smoothed cross entropy is as follows:

$$H\left(\mathbf{y}_S^{LS}, \mathbf{y}\right) = \sum_{m=1}^{M} -\mathbf{y}_S^{LS} \log(\mathbf{y}), \tag{12}$$

where $\mathbf{y}$ is the one-hot encoded form of real label.

## 4. Experiment

In this section, experiments are carried out to demonstrate the efficacy of the proposed HCFPN. Initially, the datasets and evaluation metrics that are used to validate the proposed HCFPN are described in detail. Then, the comprehensive parameter configuration is presented. Finally, the results of comparative experiments utilizing various state-of-the-art approaches, as well as an ablation study, are shown to demonstrate the performance of the proposed approach and the reasons for its improvement.

### 4.1. Experiment Datasets

After screening, two well-known datasets with imbalanced category samples, which are extensively used to assess RSSC tasks, are selected for our experiments. These datasets are AID [55] and WH-MAVS [56]. Detailed descriptions are as follows:

- AID is an aerial scene classification dataset released by Wuhan University, consisting of a total of 10,000 images. It contains 30 categories of scene images, of which each category has approximately 220–420 images. Furthermore, the pixel size of each image is approximately $600 \times 600$, with spatial resolution varying from approximately 0.5 m to 8 m. Figure 4 presents a sample of each class in this dataset. Figure 4 displays several samples from this dataset.
- WH-MAVS is a multi-task and multi-temporal dataset with dual phase, 2014 and 2016. It is based on Google Earth's large-scale mosaic RGB images with the spatial size of $47,537 \times 38,100$ pixels. The dataset spans 2608 km$^2$ of the major city of Wuhan, Hubei Province, China. It comprises 23,567 labeled patch pairings with one-to-one geographical correlation between 2014 and 2016. Each patch pairing is $200 \times 200$ pixels in size and has a spatial resolution of 1.2 m. The WH-MAVS comprises 14 categories, as follows: commercial, water, administration, agricultural, transportation, industrial, green space, residential 1, residential 2, residential 3, road, bare land, parking lot, and playground. The total number of samples in each category ranged from 126 to 5850, showing that the dataset has a substantial sample unbalanced issue. For RSSC, the data from the 2016 time phase are used to conduct the experiment. Figure 5 displays several samples from this dataset.

### 4.2. Dataset Settings and Evaluation Metrics

We repeat the experiments three times to acquire reliable experimental results by randomly picking the training/testing samples. Subsequently, more convincing average results and standard deviations are provided. The training–test sample ratio of AID and WH-MAVS is divided into 5–95%, 10–90%, 20–80%, and 50–50%. To assess the classification performance, two frequently used assessment metrics, namely, overall accuracy (OA) and confusion matrix (CM), are selected. The OA is computed by dividing the overall quantity of correctly categorized images by the overall quantity of test images, allowing the RSSC model's performance to be accurately evaluated. CM is a tabular summary of the number of accurate and inaccurate predictions made by the classifier, and it is also the most basic and intuitive metric of the RSSC model's accuracy.

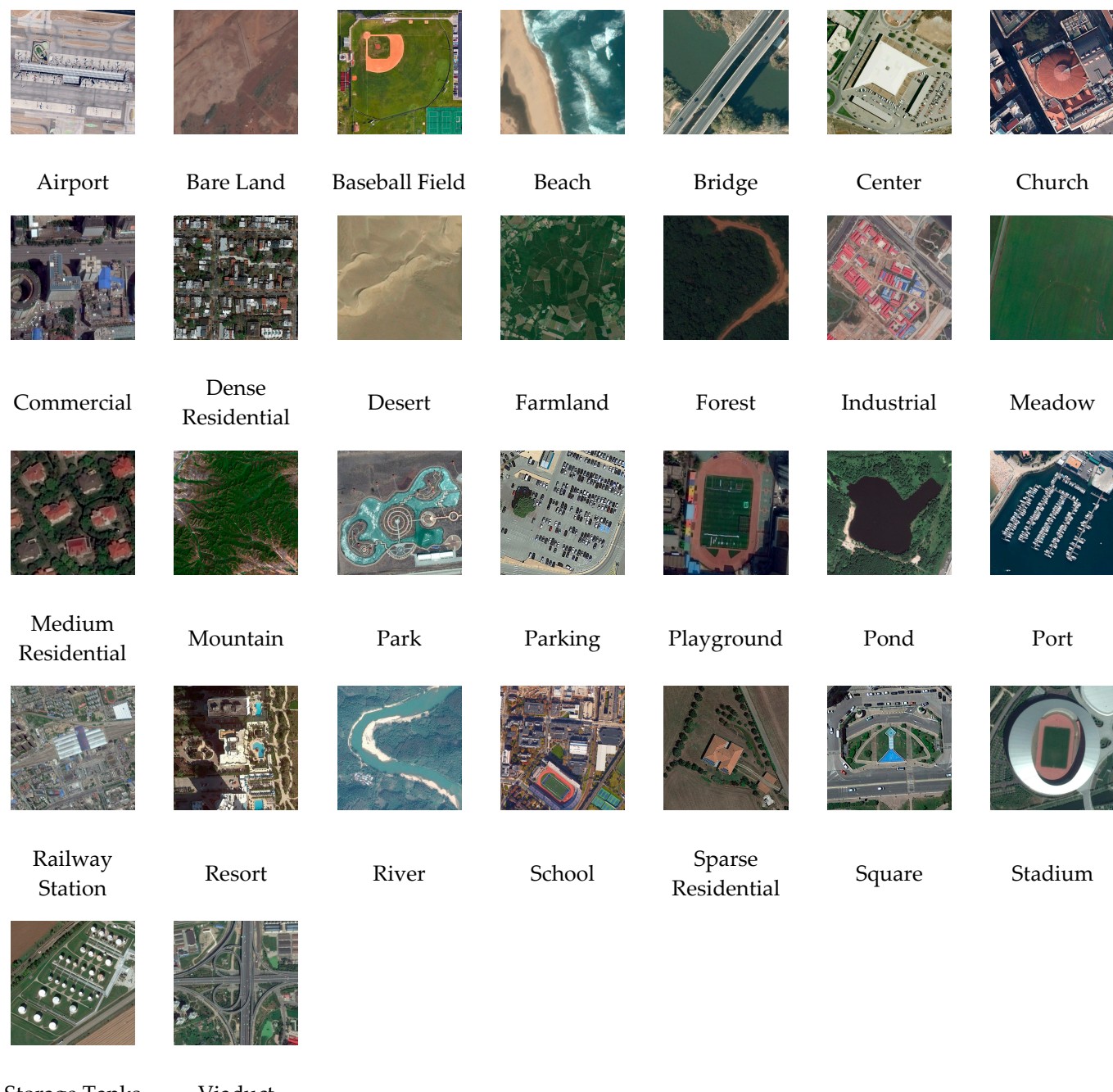

**Figure 4.** Examples of the AID dataset.

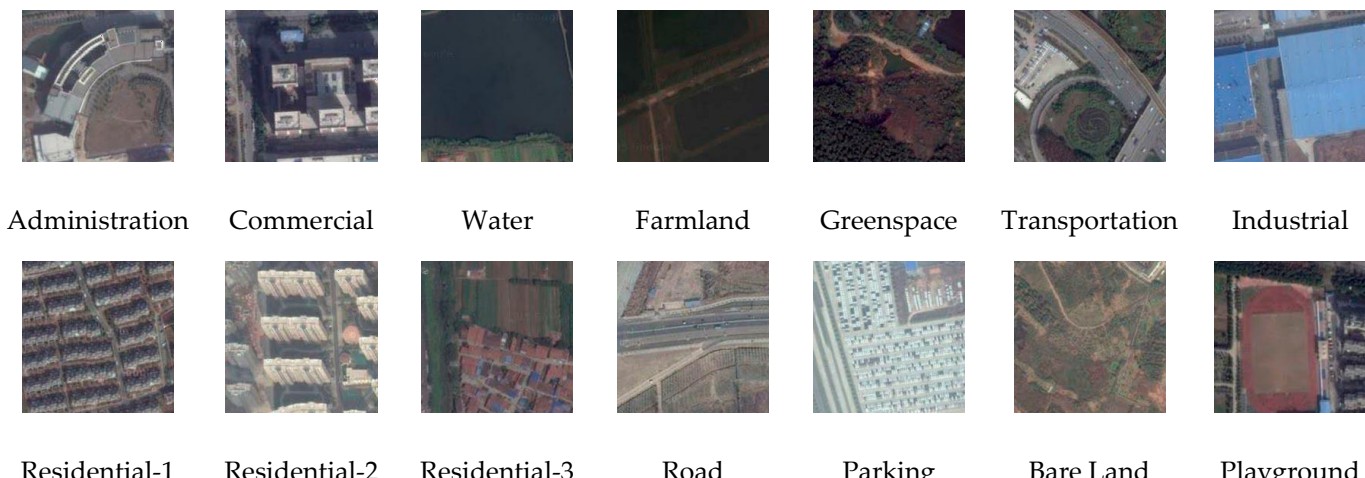

**Figure 5.** Examples of the WH-MAVS dataset.

### 4.3. Experimental Settings

Training Parameter: all the experiments are implemented using PyTorch on the GPU server with 4 Nvidia Tesla V100 and 16 GB memory. We utilize the pretrained parameters (using the ImageNet-1K dataset [57]) to initialize the ResNet34 of the HCFPN. Moreover, the remaining part of the HCFPN is initialized randomly. The SGD optimizer is selected to optimize our model for 120 epochs, with setting learning rate, momentum, and weight decay of 0.02, 0.9, and 0.0001, respectively. We also adopt a cosine decay learning rate schedule with a linear warm-up and a warm-up epoch of 5. The batch size for the distributed data parallel training framework is set to 64, and the size of the input scenes is adjusted to $224 \times 224$. In addition, we select two data augmentation methods, namely the random horizontal flipping and random erase. The size of the embedded patches of the ResNet34 layer1 and layer2 output features is $4 \times 4$ and $2 \times 2$, respectively. Finally, the smoothing parameter of the model is set to 0.1.

### 4.4. Experimental Results

In this section, the experimental results of our proposed method and some other advanced comparative methods are presented. In all these experiments, OA and CM are the evaluation metrics. Every comparison result is discussed and analyzed in accordance with various datasets.

- Results on AID dataset: Comparative experiments using different advanced approaches in RSSC are conducted. The results of the comparative experiments using four proportions of training samples are displayed in Tables 1 and 3. The proportions of the training set for comparative experiments are set to 5%, 10%, 20%, and 50%. ACNet [47], ARCNet [45], FACNN [38], and EAM [46] are utilized to compare with the proposed HCFPN. Among all the advanced approaches, the proposed HCFPN performs best in all the proportions of the training set. ACNet and ARCNet perform incredibly poorly with a few training examples. Moreover, our proposed algorithm outperforms the ACNet and ARCNet by 50.95% and 11.54%, respectively, in the case of 5% training sample ratio. Furthermore, Figure 6 shows the CM of our HCFPN on the AID dataset using 50% training samples. The classification accuracy of 21 out of the 30 categories is over 98%. These encouraging outcomes are yet another demonstration of the potency of our proposed approach.
- WH-MAVS dataset: Comparative experiments using different advanced approaches in RSSC are conducted. The results of the comparative experiments using four proportions of training samples are displayed in Tables 2 and 4. The proportions of the training set for comparative experiments are set to 5%, 10%, 20%, and 50%. ACNet [47], ARCNet [45], FACNN [38], and EAM [46] are utilized to compare with the proposed

HCFPN. Among all the advanced approaches, our HCFPN still outperforms other comparison methods, but the performance improvement of our proposed HCFPN is limited relative to the second-best method. When the proportion of the training samples is less than 5%, the accuracy of our HCFPN is still significantly higher than that of the ACNet and ARCNet by 6.75% and 3.28%, respectively.

**Table 1.** Overall accuracy (%) of HCFPN and the comparison methods when trained on the AID dataset using a proportion of 5% and 10% of the samples.

| | OA (%) | |
|---|---|---|
| **Methods** | **Training with 5% Samples** | **Training with 10% Samples** |
| ACNet [47] | 31.68 ± 6.75 | 55.80 ± 4.98 |
| ARCNet [45] | 71.09 ± 0.70 | 79.34 ± 1.02 |
| FACNN [38] | 82.00 ± 0.16 | 87.21 ± 0.32 |
| EAM [46] | 81.28 ± 1.49 | 88.17 ± 0.41 |
| HCFPN (Ours) | **82.63 ± 0.93** | **89.16 ± 0.56** |

**Table 2.** Overall accuracy (%) of HCFPN and the comparison methods when trained on the WH-MAVS dataset using a proportion of 5% and 10% of the samples.

| | OA (%) | |
|---|---|---|
| **Methods** | **Training with 5% Samples** | **Training with 10% Samples** |
| ACNet [47] | 83.45 ± 0.75 | 86.05 ± 0.78 |
| ARCNet [45] | 86.92 ± 0.25 | 90.39 ± 0.29 |
| FACNN [38] | 89.90 ± 0.29 | 91.42 ± 0.40 |
| EAM [46] | 90.17 ± 0.30 | 92.22 ± 0.13 |
| HCFPN (Ours) | **90.20 ± 0.07** | **92.33 ± 0.17** |

**Table 3.** Overall accuracy (%) of the HCFPN and the comparison methods when trained on the AID dataset using a proportion of 20% and 50% of the samples.

| | OA (%) | |
|---|---|---|
| **Methods** | **Training with 20% Samples** | **Training with 50% Samples** |
| ACNet [47] | 79.40 ± 0.56 | 89.03 ± 0.27 |
| ARCNet [45] | 87.31 ± 0.78 | 91.92 ± 0.84 |
| FACNN [38] | 91.56 ± 0.20 | 94.60 ± 0.33 |
| EAM [46] | 92.24 ± 0.32 | 95.25 ± 0.40 |
| HCFPN (Ours) | **93.04 ± 0.20** | **96.02 ± 0.24** |

**Table 4.** Overall accuracy (%) of HCFPN and the comparison methods when trained on the WH-MAVS dataset using a proportion of 20% and 50% of the samples.

| | OA (%) | |
|---|---|---|
| **Methods** | **Training with 20% Samples** | **Training with 50% Samples** |
| ACNet [47] | 90.40 ± 0.20 | 91.20 ± 0.74 |
| ARCNet [45] | 91.91 ± 1.32 | 94.28 ± 0.11 |
| FACNN [38] | 93.04 ± 0.14 | 94.24 ± 0.09 |
| EAM [46] | 93.50 ± 0.18 | 94.56 ± 0.42 |
| HCFPN (Ours) | **93.57 ± 0.21** | **94.60 ± 0.03** |

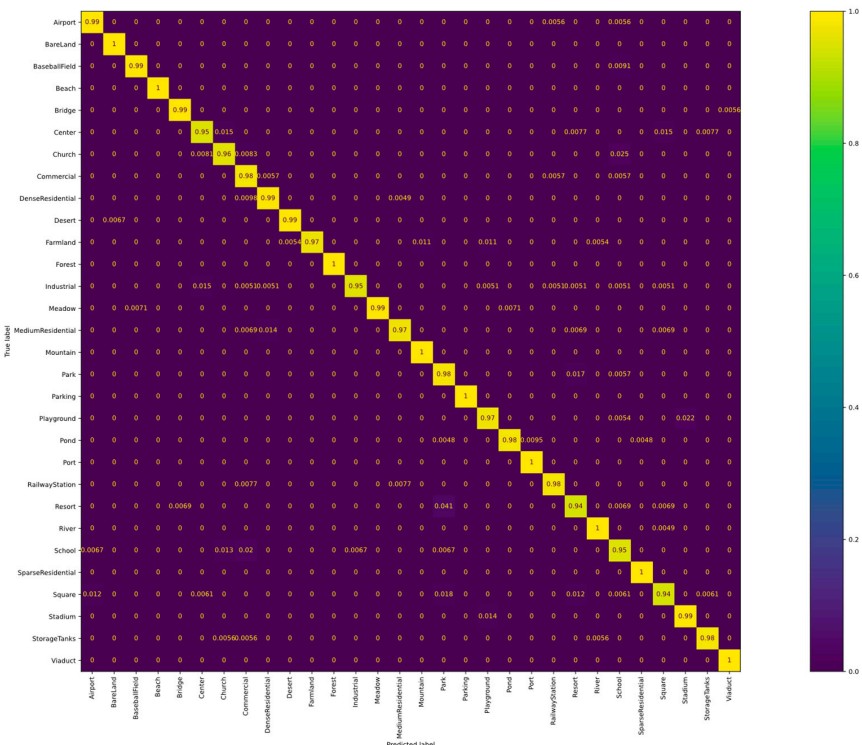

**Figure 6.** Confusion matrix of HCFPN for the AID dataset with 50% training data.

Furthermore, Figure 7 displays the CM of our proposed method on the WH-MAVS dataset utilizing 50% training samples. Ten of the 14 categories were classified with over 90% accuracy. This result proves the relatively good performance of the proposed HCFPN. Nevertheless, HCFPN performs poorly in some categories, with high inter-class similarity, such as "commercial" and "administration.".

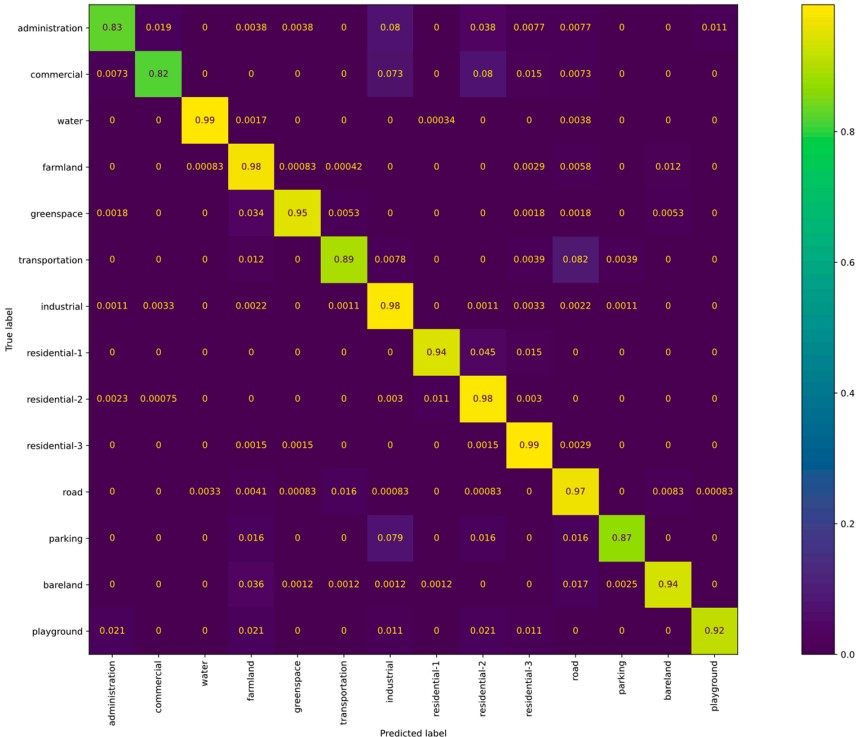

**Figure 7.** Confusion matrix of HCFPN for WHMAVS dataset with 50% training data.

*4.5. Ablation Study*

In this section, we construct the ablation experiments using the WH-MAVS dataset with 50% training sets to assess the efficacy of different components within our proposed HCFPN.

Each line in Table 5 denotes a mixture of several components, and ✓ indicates that the relevant component is used in that model. In the ablation study, model 1 performs least effectively. From the results of model 2, the label smoothing technique improves the accuracy of model 1 by 0.35%. By adding the contextual feature preserved module, the OA increases from 91.37% to 92.04%, confirming the importance of the contextual preserved module. The primary distinction between models 3 and 4 is the loss function. The classification accuracy of model 4 surpasses model 3 by 0.29%, showing that label smoothing can improve the feature representation discrimination even further.

**Table 5.** Overall accuracy (%) of HCFPN using different components when trained on the WH-MAVS dataset using a proportion of 10% of the samples.

|   | ResNet34 | Contextual Feature Preserved Module | Label Smoothing | OA (%) |
|---|---|---|---|---|
| 1 | ✓ |   |   | 91.02 $\pm$ 0.15 |
| 2 | ✓ |   | ✓ | 91.37 $\pm$ 0.18 |
| 3 | ✓ | ✓ |   | 92.04 $\pm$ 0.19 |
| 4 | ✓ | ✓ | ✓ | **92.33 $\pm$ 0.17** |

*4.6. Computational and Time Complexity Analysis*

Table 6 shows the number of floating point operations (FLOPs) for all the compared approaches. It is worth noting that HCFPN has moderate computational complexity. Compared to EAM, which has the second-best classification performance, the FLOPS of HCFPN is 6.8% lower. In addition, we also counted the time spent by all algorithms with 90% of the testing samples, as shows in Table 7. From the statistical results, the time efficiency of HCFPN is reasonable in both cases, and its added contextual feature preserved module does not have a significant negative impact on the time efficiency of the model. The additional time spent by HCFPN in extracting global long-term contextual features is worthwhile in terms of the improvement in classification accuracy.

**Table 6.** The number of FLOPs for all models when an image of size $3 \times 224 \times 224$ is input.

|   | ACNet | ARCNet | FACNN | EAM | HCFPN |
|---|---|---|---|---|---|
| FLOPs | $15.37 \times 10^9$ | $1.86 \times 10^9$ | $15.68 \times 10^9$ | $7.69 \times 10^9$ | $7.16 \times 10^9$ |

**Table 7.** Time statistics (in seconds) required to test models with HCFPN and other advanced comparison algorithms under 90% testing samples.

|   | ACNet | ARCNet | FACNN | EAM | HCFPN |
|---|---|---|---|---|---|
| AID | 1.167 | 0.545 | 0.554 | 0.601 | 1.091 |
| WH-MAVS | 0.499 | 0.207 | 0.220 | 0.284 | 0.385 |

**5. Discussion**

In this paper, we have proposed HCFPN, which combines the advantages of CNN and transformer, considering the global contextual information and local neighborhood information. Moreover, the HCFPN achieves superior performance compared to other advanced methods on two public benchmarks.

The above quantitative analysis shows that our method can still achieve better performance with a smaller proportion of training samples, indicating that our proposed HCFPN has a stable robustness. By displaying the confusion matrix, we discover that,

on the AID dataset, HCFPN can accurately classify the majority of the classes. On the WH-MAVS dataset, our proposed method performs effectively on most of the classes but poorly on a few. This finding may be related to the very similar sample distribution between classes. By additional analysis of ablation experiments, the HCFPN can extract multilevel global long-term contextual information, and it can completely merge with the high-level semantic information retrieved by CNN, achieving more discriminative features. Moreover, the label smoothing technique is used to improve the classification performance of our proposed HCFPN. As well as illustrating the performance and effectiveness of the HCFPN, we also analyze the computational complexity and time complexity of HCFPN. From the above analyzed results, our proposed method is comparative with other state-of-the-art approaches. The small sacrifice in computational complexity and time complexity of HCFPN is worthwhile compared to the significant improvement in classification accuracy.

This manuscript has demonstrated that combining the advantages of CNNs and ViT is feasible for RSSC. The multilevel convolutional features and high-level semantic features can be extracted by CNNs, while the ViT are able to effectively capture global long-term contextual features.

## 6. Conclusions

In this manuscript, we propose an HCFPN for HRRS scene classification. Different from the traditional CNN-based methods, HCFPN utilizes the transformer block to focus on self-attention and cross-level attention to enhance the model's capacity to extract long-term global contextual information. Meanwhile, the local high-level semantic features and long-term global contextual features may be extremely effectively aggregated using the category score average module. Moreover, label smoothing further enhances the robustness of the model and avoids model overfitting. Finally, the proposed HCFPN has obtained a rather acceptable state-of-the-art performance, as shown by extensive experiments on several difficult remote sensing scene classification datasets.

**Author Contributions:** Methodology, J.Y.; Writing—original draft, J.Y.; Project administration, S.W. All authors have read and agreed to the published version of the manuscript.

**Funding:** This work was supported by the National Natural Science Foundation of China Major Program under Grants 42192580 and 42192583.

**Data Availability Statement:** Data used in this paper can be provided by Jingwen Yuan (jingwenyuan@whu.edu.cn) upon request.

**Acknowledgments:** The authors would like to thank the handing editor and all anonymous reviewers for providing insightful and helpful comments.

**Conflicts of Interest:** The authors declare no conflict of interest.

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
