# Peer review of "HCFPN: Hierarchical Contextual Feature-Preserved Network for Remote Sensing Scene Classification"

_remotesensing, doi:10.3390/rs15030810_

Round 1

Reviewer 1 Report

The authors have proposed a hierarchical contextual feature-preserved network for remote sensing scene classification. The manuscript is complete, and the authors try to prove the progressiveness of the algorithm through experiments. However, there are some problems that need to be revised. The comments are as follows

1.      First, the computational complexity of the algorithm needs to be analyzed and compared with SOTA algorithm.

2.      Please explain the significance of remote sensing classification mentioned in the text? For example, how to distinguish between school scenes and playground scenes. This is a difficult problem. It seems that the algorithms proposed in this paper cannot be distinguished. In addition, the proposed classification is no different from the general natural image classification. This is the biggest problem of this paper.

3.      Thirdly, the references used in the paper are relatively old, so it is recommended to update them. In addition, some more methods regarding remote sensing using graph-based methods should be investigated in your introduction, e.g., Semi-Supervised Locality Preserving Dense Graph Neural Network With ARMA Filters and Context-Aware Learning, Unsupervised Self-correlated Learning Smoothy Enhanced Locality Preserving Graph Convolution Embedding Clustering, Self-supervised Locality Preserving Low-pass Graph Convolutional Embedding, AF2GNN: Graph Convolution with Adaptive Filters and Aggregators, Multi-feature Fusion: Graph Neural Network and CNN Combining.

4.      Fourthly, how about the adaptability of the algorithm to different number of training labels, especially small labels. Please compare with the SOAT methods.

5.      What is the adaptability of the algorithm proposed by the authors to image noise? Please use experiments to prove the progressiveness of the algorithm. That is to say, what is the classification performance of the algorithm when images are injected with different noises. In addition, when the classes are unbalanced, what is the classification effect of the algorithm.

6.       Finally, I don't think that just combining CNN and VIT to classify images is enough to show the innovation of the paper.

Author Response

We would like to take this opportunity to gratefully thank the reviewer for his/her constructive comments and recommendations for improving the paper. A point-by-point, item-by-item response to the interesting comments raised by the reviewer follows. Please see the attachment.

Reviewer 2 Report

Please see the comments.pdf for details.

Author Response

(The authors gave the same response as above.)

Reviewer 3 Report

This paper presents hierarchical contextual feature-preserved network to obtain better performance for scene classification. The final experimental results evaluate the proposed algorithm outperforms most state-of-the-art approaches. The paper is well organized and meaningful in real applications. Before being published in this journal, the following comments need to be considered.

1. I think it is promising in improving the performance of scene classification based on the thought of hierarchical classification. Moreover, the authors are recommended to further describe the second contribution how to merge the correlations of multilevel convolutional features in more detail.

2. Figure 1: What do the series of yellow arrows mean? It is better to be explained in the illustration.

3. Figure 2: 1) What do the abbreviations (i.e., “KV”, “Q”, “F1” and “F2”) mean? Suggest to supplement the description of abbreviations in the figure; 2) The description of abbreviations (i.e., “HTD”) should be explained at first use.

4. Figure 3: The content of figure should be kept in step with the illustration of figure. For example, “HTB uses Multiheaded Self-attention (MSA) and Multilayer Perceptron (MLP) Blocks in alternating layers”. Why do you describe MSA, while MSA doesn’t appear in the figure? Please recheck all the figures carefully.

5. Page 7, Line 248: What is the meaning of Ki? Ki is equal to B2Wv in Eq. (9)? Why the definition of dk is not same? Please check and correct them.

6. Page 8, Line 284: Could you explain how to set the smoothing parameter alpha? According to Line 340 on Page 10, the smoothing parameter of the model is set to 0.1. What is the theoretical or experimental basis?

7. Page 13, Line 398: The authors are suggested to expand the content of discussion. This section is very important for readers. Besides, the time optimization problem of the proposed method is encouraged to be further discussed.

Author Response

(The authors gave the same response as above.)

Round 2

Reviewer 1 Report

No more comments.